# Could 18-FDG PET-CT Radiomic Features Predict the Locoregional Progression-Free Survival in Inoperable or Unresectable Oesophageal Cancer?

**DOI:** 10.3390/cancers14164043

**Published:** 2022-08-22

**Authors:** Berardino De Bari, Loriane Lefevre, Julie Henriques, Roberto Gatta, Antoine Falcoz, Pierre Mathieu, Christophe Borg, Nicola Dinapoli, Hatem Boulahdour, Luca Boldrini, Vincenzo Valentini, Dewi Vernerey

**Affiliations:** 1Radiation Oncology Department, Neuchâtel Hospital Network, CH-2300 La Chaux-de-Fonds, Switzerland; 2Radiation Oncology Department, University Hospital of Besançon, F-25000 Besancon, France; 3Radiation Oncology Department, Centre Eugène Marquis, F-35042 Rennes, France; 4INSERM, Établissement Français du Sang Bourgogne Franche-Comté, UMR1098, Interactions Hôte-Greffon-Tumeur/Ingénierie Cellulaire et Génique, Bourgogne Franche-Comté University, F-25000 Besancon, France; 5Methodology and Quality of Life Unit in Oncology, University Hospital of Besançon, F-25000 Besancon, France; 6Dipartimento di Scienze Cliniche e Sperimentali, Università degli Studi di Brescia, I-25123 Brescia, Italy; 7Department of Digestive Surgery and Liver Transplantation, University Hospital of Besançon, F-25000 Besancon, France; 8Department of Medical Oncology, University Hospital of Besançon, F-25000 Besancon, France; 9Dipartimento di Diagnostica per Immagini, Radioterapia Oncologica ed Ematologia, Fondazione Policlinico Universitario “A. Gemelli” IRCCS, I-00100 Rome, Italy; 10EA 4662-“Nanomedicine Lab, Imagery and Therapeutics”, Nuclear Medicine Department, University Hospital of Besançon, F-25000 Besancon, France

**Keywords:** oesophageal cancer, chemoradiotherapy, neoadjuvant treatment, radiomics, predictive factors, 18-FDG PET-CT

## Abstract

**Simple Summary:**

Almost 20% of patients with a locally advanced oesophageal cancer presented long-term local control after exclusive chemoradiotherapy (CRT) and could potentially avoid surgery and its morbidity and mortality. With the aim of identifying the patients that would present long-term locoregional control, we analysed in this study the potential of some pre-treatment positron-emission tomography-computed tomography (PET-CT)-based features in predicting long-term responses and local control in patients treated with definitive CRT. Our results show that radiomics allows for the identification of patients with long-term locoregional control. If confirmed on larger populations, our results could allow the identification of patients who are good responders to CRT, and who could potentially avoid surgery.

**Abstract:**

Background: We evaluated the value of pre-treatment positron-emission tomography–computed tomography (PET-CT)-based radiomic features in predicting the locoregional progression-free survival (LR-PFS) of patients with inoperable or unresectable oesophageal cancer. Material and Methods: Forty-six patients were included and 230 radiomic parameters were extracted. After a principal component analysis (PCA), we identified the more robust radiomic parameters, and we used them to develop a heatmap. Finally, we correlated these radiomic features with LR-PFS. Results: The median follow-up time was 17 months. The two-year LR-PFS and PFS rates were 35.9% (95% CI: 18.9–53.3) and 21.6% (95%CI: 10.0–36.2), respectively. After the correlation analysis, we identified 55 radiomic parameters that were included in the heatmap. According to the results of the hierarchical clustering, we identified two groups of patients presenting statistically different median LR-PFSs (22.8 months vs. 9.9 months; HR = 2.64; 95% CI 0.97–7.15; *p* = 0.0573). We also identified two radiomic features (“F_rlm_rl_entr_per” and “F_rlm_2_5D_rl_entr”) significantly associated with LR-PFS. Patients expressing a “F_rlm_2_5D_rl_entr” of <3.3 had a better median LR- PFS (29.4 months vs. 8.2 months; *p* = 0.0343). Patients presenting a “F_rlm_rl_entr_per” of <4.7 had a better median LR-PFS (50.4 months vs. 9.9 months; *p* = 0.0132). Conclusion: We identified two radiomic signatures associated with a lower risk of locoregional relapse after CRT.

## 1. Introduction

Worldwide, oesophageal cancer is the eighth most frequent malignant disease and the sixth most prevalent cause of cancer-associated deaths, with an estimated number of 20,640 new cases and 16,410 deaths in the US in 2022 [1]. Prognosis is strongly associated with the stage at diagnosis. Most patients are diagnosed at a locally advanced stage, and the 5-year survival rate is low (13.9% in the EUROCARE-5 study) [2]. Accurate staging at diagnosis is important to define the best therapeutic strategy [3].

The standard treatment for patients with locally advanced oesophageal cancer is preoperative chemoradiotherapy (CRT), followed by surgery for operable patients [3,4,5,6,7], and exclusive CRT for inoperable patients, patients with unresectable tumours, or patients refusing surgery [3,8,9,10]. Nevertheless, preoperative CRT can obtain a pathological complete response (pCR) in almost 30% of patients, as has been shown in the CROSS study (23% obtained a pCR in adenocarcinoma and 49% did in squamous cell carcinoma) [4,5]. Those patients would have benefited from a surveillance strategy to potentially avoid the risk of surgical mortality and morbidity. Some studies have suggested that omitting surgery and adopting a “wait and see” strategy in patients presenting a complete clinical response (cCR) after CRT could be considered [10,11]. This evaluation is done at least 4 weeks after the end of CRT, and a cCR is defined as having no significant fixation in positron-emission tomography-computed tomography (PET-CT), disappearance or near-disappearance of any endoluminal tumours with negative biopsies (>4) with oesophagogastroscopy, and no progression of visible masses or thickening of the oesophageal wall with a CT scan [10,11]. However, an accurate prediction of the risk of disease evolution remains limited, thus limiting the possibility of avoiding surgery in responding patients.

Radiomics is a non-invasive, quantitative, and inexpensive approach, allowing the detection of tumour heterogeneity by extracting characteristics from medical images using data identification algorithms [12,13,14]. Some previous studies already have shown the potential of radiomic approaches in predicting the tumour biology and behaviour of bronchial, colorectal, and nasopharyngeal cancers [15,16,17]. For these reasons, there is a major interest in radiomics for radiation oncology to adapt treatments to the tumour profile and thus individualize cancer management [13,18,19].

Nevertheless, few data have been published exploring the value of radiomic approaches for oesophageal cancer patients.

The purpose of this study was to evaluate the value of radiomic features extracted from pre-treatment PET-CT in predicting locoregional progression-free survival after exclusive CRT for patients with inoperable or unresectable oesophageal cancer.

## 2. Materials and Methods

### 2.1. Population

We retrospectively analysed patients treated with CRT at the University Hospital of Besançon (France), between February 2013 and July 2019. We initially identified 137 patients, then we defined the following inclusion criteria: >18 year-old patients, histologically confirmed squamous cell carcinoma or adenocarcinoma, unresectable (locally advanced stage) or inoperable patients, PET-CT was performed before CRT, and radiotherapy treatment was administered at a total dose of 50 Gy. Based on these criteria, 91 patients were finally excluded in order to obtain the most homogeneous population for our analyses (see Figure 1). Staging was performed based on clinical, radiological, and endoscopic data according to the 7th Edition of the AJCC cancer staging manual [20]. A local ethics committee approved this retrospective study, and data were anonymously collected on a web-based data collection platform.

### 2.2. PET-CT Acquisition Technique

All patients fasted for at least 4 h before receiving an intravenous injection of 3.5 MBq/kg of fluoro-2-deoxyglucose (FDG). All acquisitions were realised on a GE DISCOVERY 690 PET-CT (GE Healthcare, Milwaukee, WI, USA), at the Nuclear Medicine Department of the University Hospital of Besançon (France). The acquisition protocol included an acquisition from the top- to the mid-thigh performed 60 min after injection, comprising 7 to 8 bed positions with an acquisition time per bed position varying between 1.5 and 2.5 min, depending on the patients’ body mass index. PET images were reconstructed using a standard iterative algorithm. A “low dose” CT scan (50 to 210 mA, 120 kV, and a section of 3.75 mm thickness) was performed for the attenuation correction of the PET data.

### 2.3. Chemoradiotherapy Schedule

All the patients received radiotherapy. Following our local protocols, 2 clinical target volumes (CTVs) were defined as follows:-CTV 1 = T and N GTV + perioesophageal interfaces and nodal drainage (depending on the tumour localisation in the oesophagus) + 5 cm in the cranio-caudal direction;-CTV 2 = T and N GTV + perioesophageal interfaces and nodal drainage (depending on the tumour localisation in the oesophagus) + 3 cm in the cranio-caudal direction.

Respective planning target volumes (PTVs), PTV1 and PTV2, were obtained by adding an isotropic 1-cm margin.

The total dose of the PTV1 was 40 Gy (2 Gy/fraction, 5 fractions/week), and the dose of the boost was 10 Gy (total dose of the PTV2 = 50 Gy, 2 Gy/fraction, 5 fractions/week), delivered using a sequential boost approach. Radiation therapy was mainly delivered using sliding window intensity-modulated radiation therapy (IMRT) or a volumetric arc technique (41/46 patients, 3D-conformal radiotherapy in 5/46). Weekly bone-based image guided radiotherapy was performed to verify the setup of the patients.

Concomitant chemotherapy was delivered to 42 patients, either using a carboplatin AUC 2 and paclitaxel 50 mg/m^2^ intravenous regimen on D1, D8, D15, D22, and D29 (5 cycles), or a FOLFOX protocol (oxaliplatin 85 mg/m^2^ on D1, folinic acid 200 mg/m^2^ on D1, 5-fluorouracil bolus 400 mg/m^2^ on D1, and continuous 5-fluorouracil 800 mg/m^2^/day on D1 and D2), 6 intravenous cycles in total every 14 days (3 cycles during radiotherapy starting at D1, and 3 cycles after radiotherapy).

### 2.4. Delineation of the Metabolic Volumes Studied on PET-CT and Extraction of Radiomic Parameters

Textual analysis was performed for each included patient. The tumour volume of interest (VOI) was defined by two radiation oncologists with pre-treatment PET-CT images of the tumour.

Image features were extracted using the MODDICOM platform, an R library able to extract 232 different image features [21,22]. MODDICOM complies with the quality standards defined by the image biomarker standardization initiative [23], a standardization initiative aiming at classifying the features by families (e.g., morphological, intensity histogram, grey level distance, etc.) and harmonizing computational methods among the different radiomic software tools. In our study, to avoid artefacts in image pre-processing, no interpolations or filters (e.g., Laplacian of Gaussian) were applied to images.

### 2.5. Statistical Methods and Endpoints

Patients’ characteristics were described with mean (standard deviation, SD), and median (interquartile range, IQR) and frequencies (percentage) for continuous and categorical variables, respectively, and compared between groups of patients using the Wilcoxon test and chi-square test (or Fisher’s exact test, if appropriate).

The primary endpoint was the analysis of the locoregional progression-free survival (LR-PFS), defined as the time between the date of diagnosis and local or regional failure or death. Patients alive without local or regional progression or lost to follow-up were censored at the date of the last contact. Patients with distant progression were censored at the date of the distant progression.

Secondary endpoints were overall survival (OS, defined as the time between the date of diagnosis and death), progression-free survival (PFS, defined as the time between the date of diagnosis and local or distant failure, second cancer, or death, whichever occurred first). Patients alive without presenting one of those events were censored at the date of the last contact.

Survival was estimated using the Kaplan–Meier method and compared using the log-rank test.

The evaluation of tumour response was based on a set of combined criteria evaluated after CRT: PET-CT, a thoraco-abdomino-pelvic CT scan, and endoscopy. The response was classified as complete response, partial response, stability, or progression.

### 2.6. Strategy for Radiomic Parameter Analysis

#### 2.6.1. Step 1: Correlation Analysis for Parameters Selection

The Pearson correlation matrix of the 230 radiomic parameters was estimated. To limit overfitting due to the limited number of patients and parameter redundancy, the number of parameters was reduced: when correlation between 2 parameters was higher than 0.9, only one was randomly selected in our analysis.

#### 2.6.2. Step 2: Hierarchical Clustering on Parameters Selected

Hierarchical clustering for (1) individuals and (2) parameters based on Euclidean distance and the Ward algorithm was used to identify groups of patients with different radiomic signatures, and a heatmap was performed for graphical representation [24]. Clinical characteristics and survivals were compared between the clusters of patients.

#### 2.6.3. Step 3: PCA on Selected Parameters

In order to find major radiomic parameters that could discretize the groups previously found, a principal component analysis (PCA) was performed. Patients were projected on the 2-principal axes, and ellipses were drawn around these groups with a 95% data concentration. The parameter contribution for the axes was assessed and the association between the parameters with the highest contribution and endpoints was estimated using the percentile corresponding to the repartition of the groups of patients. The restricted cubic spline (RCS) method was also used to investigate the relevant cut-off for association with LR-PFS.

#### 2.6.4. Step 4: Sensitivity Analysis on All Parameters

As the sensitivity analysis, a lasso regression on LR-PFS including all parameters was performed with a cross-validation procedure to optimize the penalty parameter.

## 3. Results

### 3.1. Description of the Population

Table 1 summarises the patients’ features. The male/female ratio was 35/11. Thirty-five out of forty-six patients had a 0–1 ECOG performance status, with 29/46 of them presenting weight loss at diagnosis (on average, estimated at 10% relative to body weight), dysphagia in 31/46, and asthenia in 21/46. Forty patients were smokers, and a history of alcoholism was found in 31/46 patients.

### 3.2. Radiomic Parameter Analysis

#### 3.2.1. Step 1: Correlation Analysis for Parameter Selection

Initially, 230 parameters were provided by radiomic analysis. With the threshold of 0.9 for the correlation coefficient, 55 radiomic parameters were selected for the analysis (Appendix A).

#### 3.2.2. Step 2: Hierarchical Clustering on Parameters Selected

Figure 2 shows the heatmap including the 55 radiomic parameters. Blue represents the under-expression of a radiomic variable in a patient and red represents its overexpression compared with the mean value. The heatmap identified two groups of patients with different expressions of radiomic features: for the patients in group 1 (*n* = 13) the first bloc of 29 parameters was under-expressed (blue colour), and the second bloc of 26 parameters was overexpressed (red colour); for the patients in group 2 (*n* = 33), we observed the opposite.

The baseline characteristics of the two groups were universally well-balanced, except for the t-stage, where group 1 had more T1–2 tumours (62% vs. 15%; *p* = 0.0033), and the presence of a previous history of alcoholism was more frequent in group 1 (92% vs. 59%; *p* = 0.0679).

The patients in group 1 had a longer LR-PFS (median = 22.8 months, 95%CI = 6.5-NA) than those in group 2 (9.9 months, 95% CI = 7.8–23.1) (Figure 3). The OS and PFS were not significantly different between the two groups, but we observed a trend for a better survival rate for patients in group 1 (Appendix A).

In order to explore how the radiomic-based patient stratification compared to clinico-pathologic prognostication (based, for example, on histology and TN staging), we analysed the clinical characteristics of the two groups of patients and their potential impact on LR-PFS using a Cox univariate analysis. We found two variables were of potential significance: the initial Charlson comorbidity index (CCI), with an HR = 0.46 (95% CI 0.19–1.12; *p*-value = 0.0862) and the stage with an HR = 2.57 (95% CI 0.95–6.97; *p*-value = 0.063). As we had only 24 events, we developed a Cox multivariate analysis using “group 1 vs. group 2”, “CCI < 3 vs. CCI > 3”, and “t-stages 1–2 vs. t-stages 3–4”. Upon multivariate analysis, the CCIs and the T-stages lost their potential relevance, while the radiomic-based groups still influenced LR-PFS.

#### 3.2.3. Step 3: PCA on Selected Parameters (55 Radiomic Parameters and 46 Patients)

The first two axes of the PCA explained 53.8% of the variance of the data. Figure 4 shows the projection of the patients’ data on the factorial plane composed of the two first axes. Ellipses around the two groups defined by the hierarchical clustering were drawn and were separated on the first axis.

The two variables with the highest contribution to the first axis were “F_rlm_2_5D_rl_entr” and “F_rlm_rl_entr”. Both variables dealt with the entropy of the grey-level run length matrix (GLRLM) [23,25]. As group 1 represented 28% of the patients, the 28^th^ percentile was used as a threshold for both radiomic parameters: 4.7 and 3.3 for “F_rlm_2_5D_rl_entr” and “F_rlm_rl_entr”, respectively. Figure 5 and Figure 6 show a statistically significant association between both parameters regarding LR-PFS (*p*-values = 0.0343 and 0.0132). The cut-off suggested by RCS (Appendix A) close to the 28th percentile provided similar LR-PFS curves.

Patients presenting “F_rlm_2_5D_rl_entr” <3.3 times more often belonged to group 1, and patients presenting “F_rlm_2_5D_rl_entr” ≥3.3 times more often belonged to group 2 (Table 2); patients presenting “F_rlm_rl_entr_per” <4.7 times more often belonged to group 1, and patients presenting “F_rlm_rl_entr_per” ≥4.7 times more often belonged to group 2 (Table 3).

### 3.3. Sensitivity Analysis

By using lasso regression, three parameters were identified: “F_stat_entropy”, “F_rlm_glnu”, and “F_szm_2_5D_zs_var” [23]. “F_stat_entropy” was highly correlated with “F_rlm_rl_entr” and “F_rlm_2_5D_rl_entr” (correlation = 0.88; Table 4).

Figure 7, Figure 8, Figure 9, Figure 10 and Figure 11 show the LR-PFS curves for these three parameters using the cut-off derived from the RCS analysis. The thresholds found with this sensitivity analysis are similar with those presented above in the Figure 2, Figure 3, Figure 4 and Figure 5 (Appendix A).

## 4. Discussion

In this study, we identified two radiomic signatures allowing for the identification of patients presenting a good long-term LRFS after exclusive CRT. Our findings, if confirmed on a larger, independent population of oesophageal cancer patients, could be a way to identify patients presenting a good prognosis, which could prevent local surgery.

Our study had some limitations. Firstly, the absence of histological evidence of a complete response to CRT: the histological proof of a pCR is more robust than the definition of a cCR that we used in our study that was based on clinical and radiological criteria. Even if a “wait and see” policy has been proposed, the pathological data from patients who received a scheduled oesophagectomy following CRT demonstrated that the cCR does not always accurately reflect a pathological CR [26].

Another limit was the limited number of patients finally included in this study. Many patients initially identified were finally excluded, as we preferred to have a homogeneous population and reduce all the biases that could influence the results. For example, we excluded patients who underwent a PET-CT scan in another centre, as this is a known potential bias in radiomic studies. From a methodological point of view, our model would require internal cross-validation on the same population or an independent validation in a larger cohort of patients to confirm the predictive performance of our model. However, these limits are often found in radiomic studies when these studies deal with rare conditions (such as oesophageal cancer).

Nevertheless, we will discuss and comment on the data. Currently, the question of surgery for oesophageal cancer patients presenting a cCR after CRT is often an object of debate, even if surgery remains the reference treatment after CRT in operable patients. Prospective trials and meta-analyses have shown that CRT + surgery significantly improve the OS of locally advanced oesophageal carcinoma patients when compared with surgery alone [4,5,27,28]. Nevertheless, some studies have shown that patient survival could be identical after CRT in both operated and non-operated patients, but the locoregional relapse rates were higher without surgery [28,29]. It was reported that the cCR after CRT is a positive predictive factor [30], and some studies have shown that there was no difference in terms of 5-year OS and disease-specific survival (DSS) between patients receiving neoadjuvant CRT followed by surgery and those receiving definitive CRT achieving a cCR [26,30,31]. Nevertheless, these favourable outcomes in patients presenting a cCR have not been confirmed in all the available studies [32,33]. With this context, surveillance could be utilized in patients presenting a cCR and high surgical risks [11,26]. The challenge would therefore be to identify them from the outset in order to avoid surgery-related morbidity and mortality.

Our results showed that the radiomic approach has a discriminating and predicting value in the initial evaluation of the potential clinical behaviour of oesophageal cancers. In our study, 55 radiomic parameters allowed us to classify our patients into two distinct groups at prognosis. The analysis of locoregional relapse on the whole population showed that there was a statistically significant difference in locoregional relapse rates between these two groups (log-rank test; *p*-value = 0.049). In our analysis, patients in group 1 were identified as “good prognosis” patients presenting lower locoregional failure rates, while patients in group 2 were identified as those presenting a “worse prognosis” and a higher risk of locoregional failure. Moreover, we also identified two radiomic parameters (“F_rlm_rl_entr_per” and “F_rlm_2_5D_rl_entr”) that were strongly correlated to LR-PFS, and we found thresholds allowing for the identification of the two groups of patients presenting different clinical results. Our model showed that the expression of the radiomic parameters, “F_rlm_rl_entr_per” < 4.7 and “F_rlm_2_5D_rl_entr” < 3.3, is correlated with a decreased risk of locoregional relapse after CRT.

If confirmed, these results could add significant supplementary criteria to the standard clinical-radiological evaluation and help to individualize and adapt the therapy to the patients’ tumour profiles.

The staging of oesophageal cancers is currently based on CT scans, positron-emission tomography-CT (PET-CT) scans, and endoscopic ultrasound (EUS). The current guidelines recommend a combination of these different modalities for staging the disease [10,34,35]. In the modern approach to oesophageal cancer patients, imaging modalities may be relevant as they give Appendix A based on the radiomic features of the tumour, thus delineating its intrinsic biological features.

These radiomic signatures could predict the clinical behaviour of the disease, and could influence the treatment approach, depending on the predicted risk of local and/or distant metastases. For example, patients with a higher risk of locoregional relapse could be eligible for more “aggressive” local therapeutic strategies (e.g., surgery and/or an increased dose of radiotherapy on the GTV in inoperable patients), and patients with lower PFS rates could be eligible for trials exploring the role of adjuvant chemotherapy. At the same time, patients presenting a better prognosis (based on the radiomic features) could be eligible for trials exploring the role of “wait and see” approaches in operable patients.

In the literature, only a few studies are available on PET-CT radiomics and oesophageal cancer. A recent systematic review by Deantonio et al., summarising the impact of 18F-FDG PET-CT-based radiomics in predicting the response to neoadjuvant chemoradiotherapy in oesophageal cancer, identified only five studies where radiomics was used to predict a pCR [36]. These authors concluded that radiomic models exhibited a good performance in predicting pathological complete responses (pCRs), with a pooled area under the curve (AUC) of 0.82 (95% CI: 0.74–0.9). The authors highlighted the great potential of 18F-FDG PET-CT-based radiomics to predict pCRs in oesophageal cancer patients receiving neoadjuvant chemoradiotherapy. Our results compared well with the available literature.

Beukinga et al. analysed 97 pre-treatment 18F-FDG PET-CT scans of consecutive patients with locally advanced oesophageal cancer, included in a prospective single-institution database [37]. All patients received neoadjuvant CRT (carboplatin/paclitaxel/41.4 Gy) followed by surgery. These authors concluded that the predictive values of the radiomic-based models were superior to the standard method (SUV_max_ in their study), and that radiomic approaches should be further evaluated to refine their predictive value in justifying the omission of surgery.

Xiong and al. studied 440 radiomic features that were extracted from both pre- and mid-CRT ^18^F-FDG PET-CT images of 30 patients: a random forest model incorporating both clinical and radiomic features achieved the best predictive performance (when compared with clinical features alone), with an accuracy of 93.3%, a specificity of 95.7%, and a sensitivity of 85.7% [38].

These studies were confirmed in a retrospective study of Chen et al. on 44 patients, showing that ^18^F-FDG PET-CT-derived radiomic information is useful for predicting the surgical pCR after CRT [39]. The authors concluded that by using a combination of clinical and radiomic parameters, it is possible to improve the stratification of patients into subgroups with intrinsically different clinical outcomes.

## 5. Conclusions

Radiomic features obtained from pre-treatment ^18^F-FDG PET-CT may provide robust overall information about the clinical behaviour of oesophageal cancer patients. These results warrant further studies exploring the integration of such information in the pre-treatment risk stratification to tailor the therapeutic approaches to oesophageal cancer patients.

## Figures and Tables

**Figure 1 cancers-14-04043-f001:**
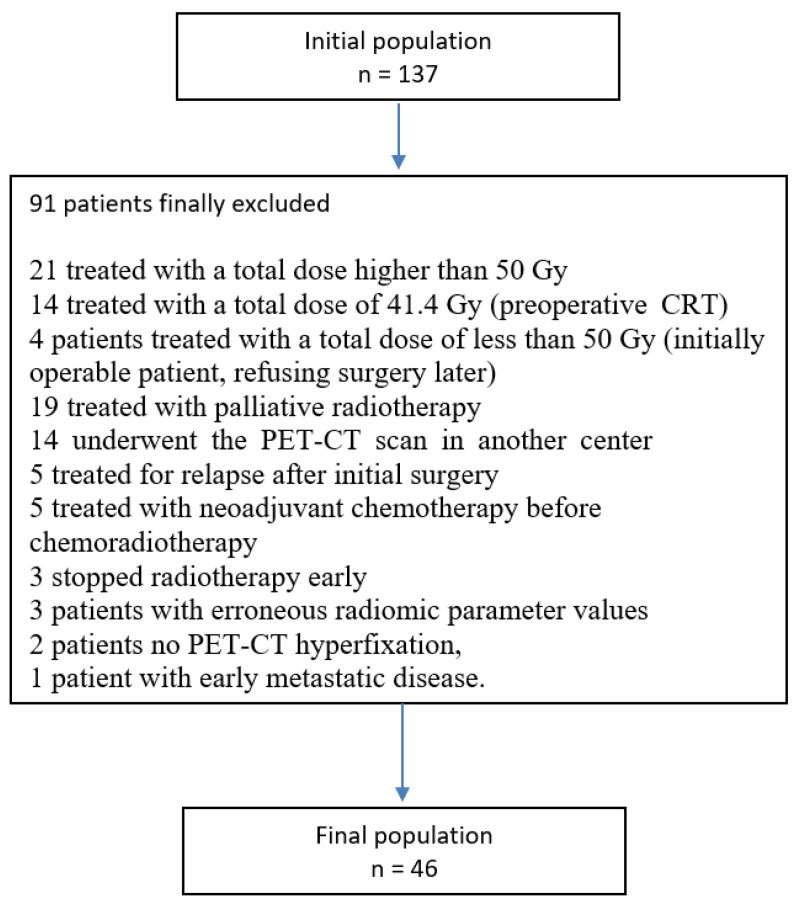
Reasons for excluding patients who were not finally included in this analysis.

**Figure 2 cancers-14-04043-f002:**
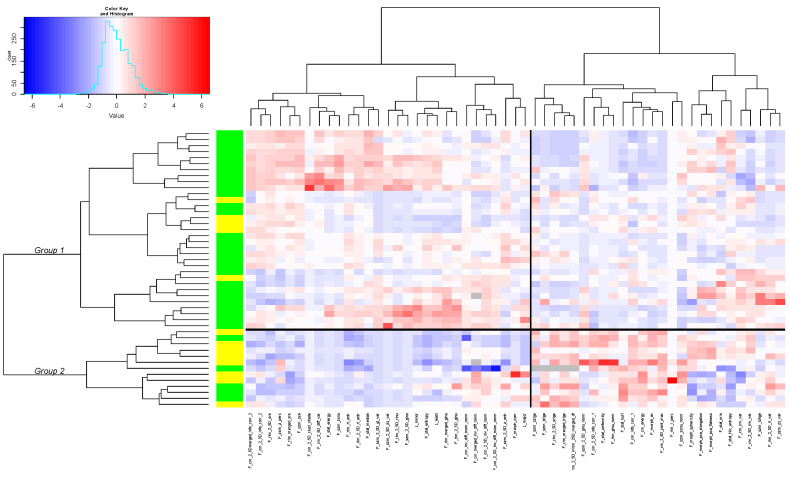
Heatmap including 55 radiomic parameters and 46 patients. Green: patients T3-4, yellow: patients T1-2.

**Figure 3 cancers-14-04043-f003:**
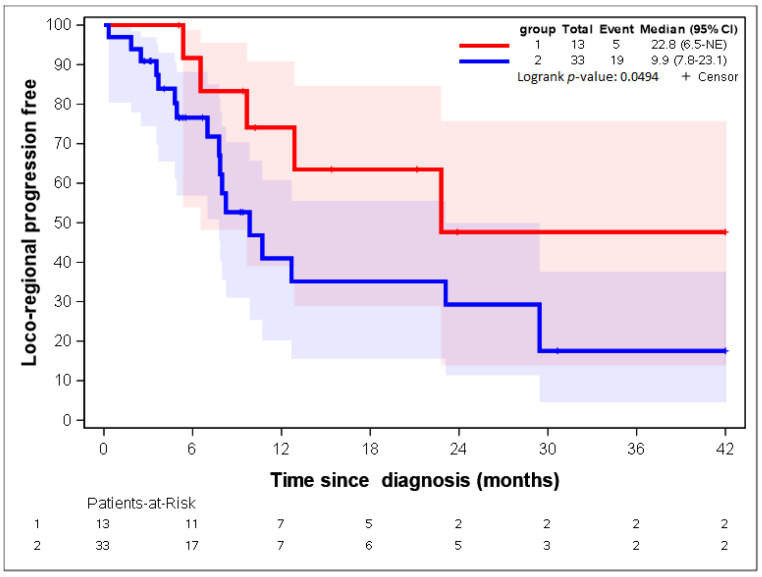
LR-PFS of the two groups of patients identified with the heatmap.

**Figure 4 cancers-14-04043-f004:**
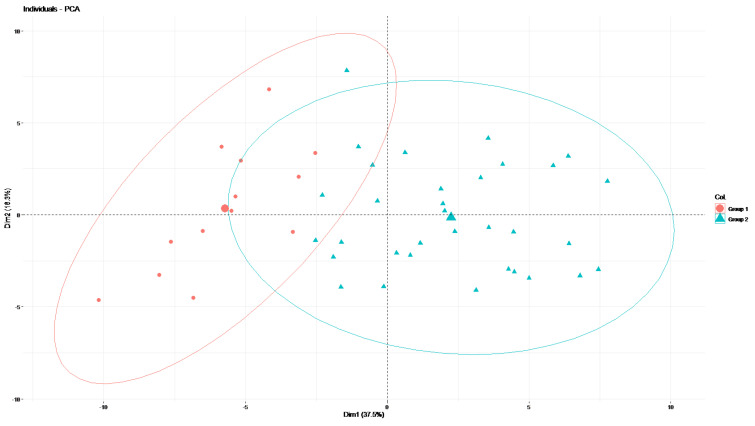
Results of the PCA.

**Figure 5 cancers-14-04043-f005:**
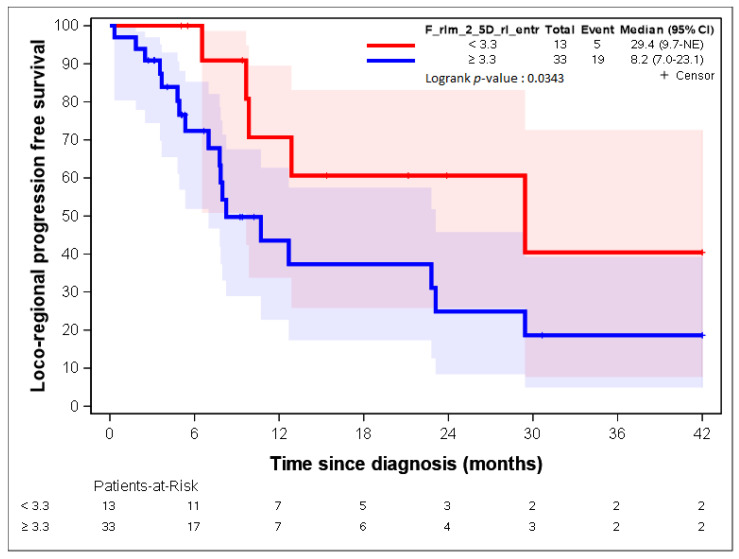
LR-PFS depending on the expression of the “F_rlm_2_5D_rl_entr” radiomic variable. See [23] for the precise definition of this variable.

**Figure 6 cancers-14-04043-f006:**
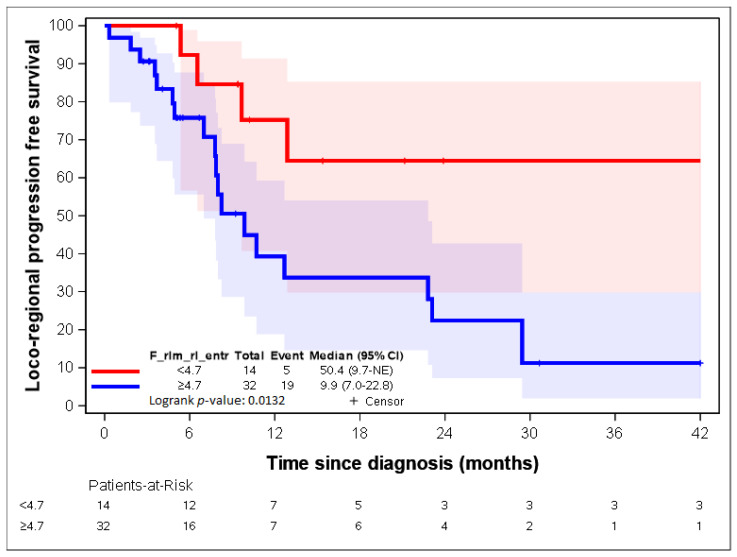
LR-RFS depending on the expression of the “F_rlm_rl_entr” radiomic variable. See [23] for the precise definition of this variable.

**Figure 7 cancers-14-04043-f007:**
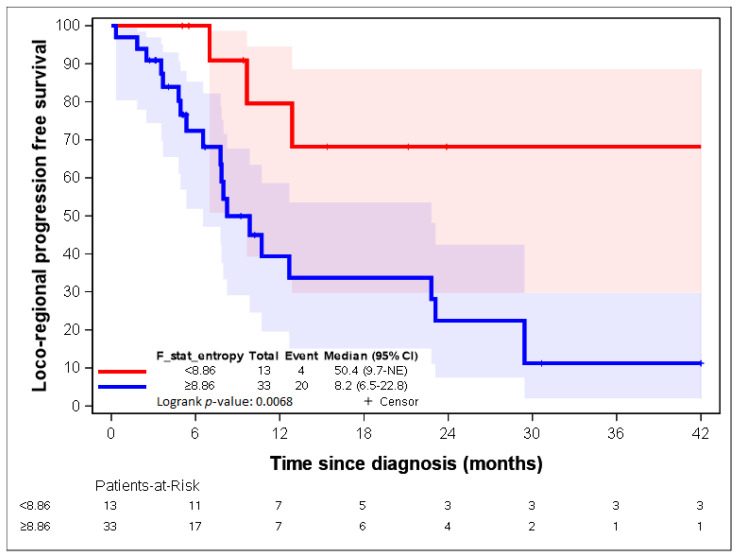
LR-RFS depending on the expression of the “F_stat_entr” radiomic variable. See [23] for the precise definition of this variable.

**Figure 8 cancers-14-04043-f008:**
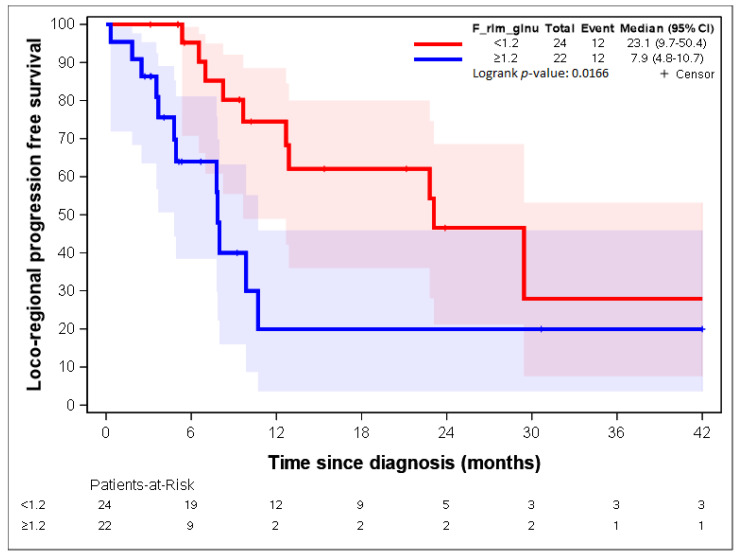
LR-RFS depending on the expression of the “F_rlm_glnu” radiomic variable. See [23] for the precise definition of this variable.

**Figure 9 cancers-14-04043-f009:**
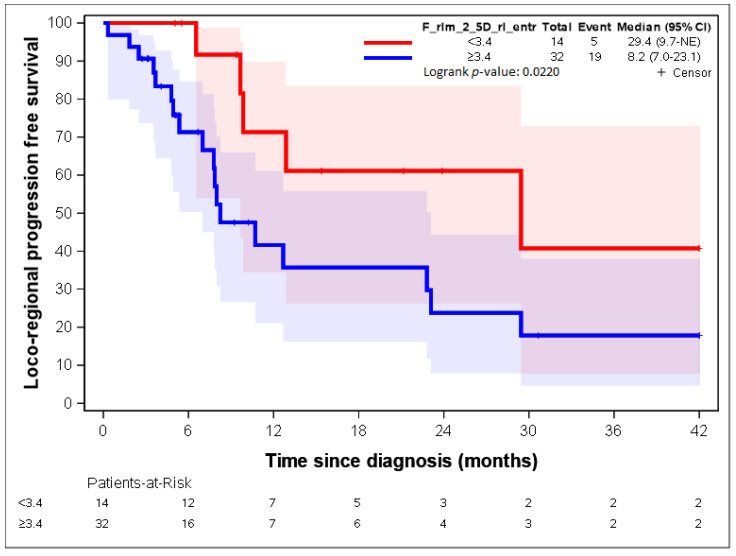
LR-RFS depending on the expression of the “F_rlm_2_5D_rl_entr” radiomic variable. See [23] for the precise definition of this variable.

**Figure 10 cancers-14-04043-f010:**
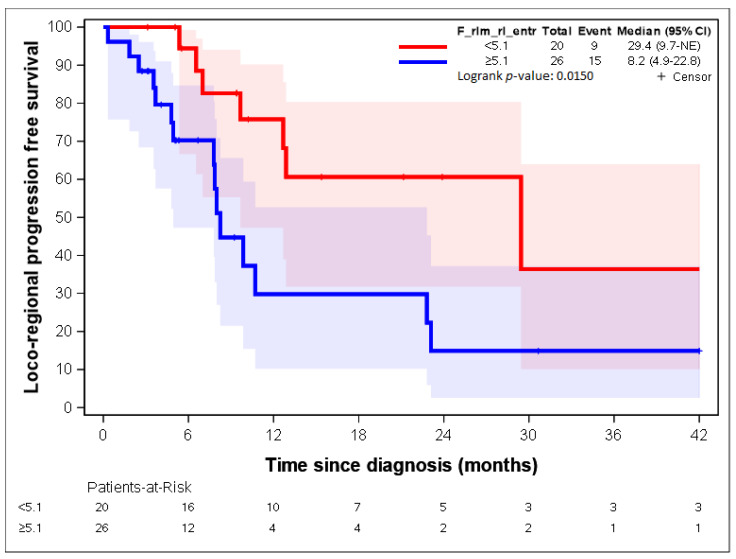
LR-PFS depending on the expression of the “F_rlm_rl_entr” radiomic variable. See [23] for the precise definition of this variable.

**Figure 11 cancers-14-04043-f011:**
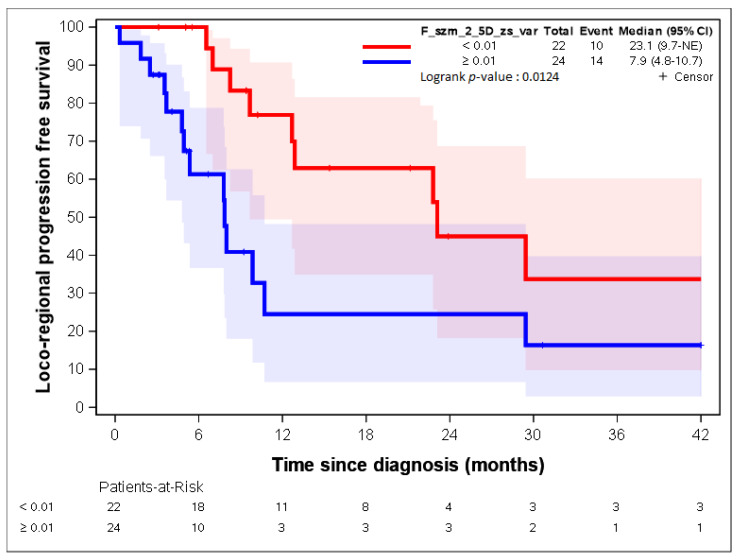
LR-PFS depending on the expression of the “F_szm_2_5D_zs_var” radiomic variable. See [23] for the precise definition of this variable.

**Table 1 cancers-14-04043-t001:** Patient and tumour features of the 46 patients enrolled in the study.

Variable	*n*	%
Whole population	46	100
Sex		
Male	35	76
Female	11	24
Age		
Median	68.8	-
Range	52.2–92.5	-
ECOG performance status		
0	13	28
1	22	48
2	10	22
3	1	2
Body mass index (BMI)		
Median	23.1	-
Range	15.4–31.9	-
<18.5	4	9
18-5–25	28	61
25–30	11	24
>30	3	6
Medical history of the patient		
History of digestive surgery		
Yes	16	35
No	30	65
History of cardio-vascular disease		
Yes	29	37
Not	17	63
History of pulmonary disease		
Yes	12	26
Not	34	74
Histology		
Squamous cell carcinoma	40	87
Adenocarcinoma	6	13
Tumour localisation		
Cervical and superior 3rd oesophageal cancer	18	39
Thoracic oesophageal cancer	17	37
Distal oesophageal cancer	7	15
Oeso-gastric junction	4	9
Tumour (T) stage		
T1	2	4
T2	11	24
T3	29	63
T4	4	9
Nodal (N)		
0	17	37
1	21	46
2	5	11
3	3	6
TNM stage		
IA	1	2
IB	5	11
IIA	11	24
IIB	5	11
IIIA	14	30
IIIB	3	7
IIIC	7	15

**Table 2 cancers-14-04043-t002:** Number of patients with the parameter “F_rlm_2_5D_rl_entr” < 3.3 or ≥ 3.3 in each patient group (chi-square test).

		Group 1	Heatmap	Group 2	
	*n*	%	*n*	%	*p*-Value
F_rlm_2_5D_rl_entr < 3.3	10	76.92	3	9.09	<0.0001
F_rlm_2_5D_rl_entr ≥ 3.3	3	23.08	30	90.91	

**Table 3 cancers-14-04043-t003:** Number of patients with the parameter “F_rlm_rl_entr_per” < 4.7 or ≥ 4.7 in each patient group (chi-square test).

		Group 1	Heatmap	Group 2	
	*n*	%	*n*	%	*p*-Value
F_rlm_rl_entr_per < 4.7	12	92.31	2	6.06	<0.0001
F_rlm_rl_entr_per ≥ 4.7	1	7.69	31	93.94	

**Table 4 cancers-14-04043-t004:** Correlation matrix.

	F_stat_entropy	F_rlm_glnu	F_szm_2_5D_zs_var	F_rlm_rl_entr	F_rlm_2_5D_rl_entr
F_stat_entropy	1	0.72	0.65	0.87	0.88
F_rlm_glnu		1	0.9	0.58	0.69
F_szm_2_5D_zs_var			1	0.52	0.63
F_rlm_rl_entr				1	0.87
F_rlm_2_5D_rl_entr					1

## Data Availability

The data presented in this study are available on request from the corresponding author. The data are not publicly available due to the respect of the privacy of the patients.

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
