# Peer review of "Could 18-FDG PET-CT Radiomic Features Predict the Locoregional Progression-Free Survival in Inoperable or Unresectable Oesophageal Cancer?"

_cancers, 2022, doi:10.3390/cancers14164043_

Round 1
Reviewer 1 Report
Reviewer comments:
Comments to the Author
This manuscript by Dr. Berardino De Bari et al., evaluated the value of pre-treatment Positron Emission Tomography–Computed Tomography (PET-CT)-based radiomic features in predicting the locoregional-progression free survival (LR-PFS) of patients with inoperable or unresectable oesophageal cancer.
The hypothesis of the manuscript is impressive, but the results are little misleading with huge errors in the Tables numbering. Authors are advised to include some suggestions.
Minor criticisms
• How the authors have decided the total dose to the PTV1 as 40Gy (2Gy/fraction, 5 Fractions/week), and the dose of the boost as 10Gy (total dose to the PTV2 = 50 Gy, 2Gy/fraction, 5 Fractions/week). Please provide the logics and references of using these doses.
• Please explain about R library to extract 232 different image features.
• Table 2, 3, 4 and Table 5 are missing or there is some typographical error.
• Please explain all the figure legends in detail including the information about number of samples used to conduct each experiment. Also, mention in the legend, which statistical analysis was performed in each figure.
• Please undergo a thorough check of the manuscript for typographical and grammatical errors.
Author Response
Comments to the Author
This manuscript by Dr. Berardino De Bari et al., evaluated the value of pre-treatment Positron Emission Tomography–Computed Tomography (PET-CT)-based radiomic features in predicting the locoregional-progression free survival (LR-PFS) of patients with inoperable or unresectable oesophageal cancer.
The hypothesis of the manuscript is impressive, but the results are little misleading with huge errors in the Tables numbering. Authors are advised to include some suggestions.
Reply : we thank the reviewer for his/her comment, appreciation and suggestions. Tables number have been reviewed and corrected
Minor criticisms
- How the authors have decided the total dose to the PTV1 as 40Gy (2Gy/fraction, 5 Fractions/week), and the dose of the boost as 10Gy (total dose to the PTV2 = 50 Gy, 2Gy/fraction, 5 Fractions/week). Please provide the logics and references of using these doses.$
Reply: it is globally based on internal protocols of treatment. In particular, our Dpt was the leading center of a randomized trial, the PRODIGE 26 trial (ClinicalTrials.gov Identifier: NCT01348217) exploring the interest of a dose escalation in inoperable esophageal cancer. In this trial, patients in the standard arm were treated with the schedule described in our article. We followed this schedule to treat our patients, as it allowed to lower the dose to the OARs by delivering 40Gy and not 50Gy on a larger volume
- Please explain about R library to extract 232 different image features.
Reply: when clicking on the URL indicated in the reference, the user could access the gitHub site (https://github.com/kbolab/moddicom). In the "Usage" section of this website, it is explained explained step by step how to extract the features for a single series, for a set of series and after applying a LoG filter.
- Table 2, 3, 4 and Table 5 are missing or there is some typographical error.
Reply: thank you very much for your correction. It is a typographical error. The numbering, now, has been corrected.
- Please explain all the figure legends in detail including the information about number of samples used to conduct each experiment. Also, mention in the legend, which statistical analysis was performed in each figure.
Reply: the number of patients is states under each figure as “Patients at risk”, as it’s usually done in scientific articles. All the methodological aspects are described in the Material and Methods section of the article, and in the figures with survival curves, it is already stated that it is a logrank test. Anyway, following the reviewer comment, in some of the figures legends we added the information about the parameters that have been analysed and the number of patients.
- Please undergo a thorough check of the manuscript for typographical and grammatical errors.
Reply: thank you for your suggestion. The article has been completely reviewed for typos, and corrected if needed.
Reviewer 2 Report
The study addresses an important question of predicting outcome following CRT for inoperable oesophageal cancer.
This is a retrospective study assessing the potential of PET radiomics to predict the primary endpoint of loco-regional progression free survival (LR-PFS).
The methods are clearly outlined and the in house radiomics extraction software has been validated in accordance with IBSI.
There are a few limitations to this study (that the authors acknowledge) - retrospective nature of the study, small sample size, high risk of overfitting given large number of features examined.
In terms of clinical relevance, I would like to understand how the radiomic based patient stratification compares to clinico-pathologic prognostication (based on histology and TN staging). It would be helpful to either compare performance of clinical model alone with a radiomic or mixed (clinico-radiomic) model to understand the added benefit of radiomic analysis or at least compare significant clinical and pathological variables between two radiomic groups to ensure that the difference in LR-PFS rates are not explained by cervical location, higher T or N stage or SCC vs adeno histology. In addition, I wonder whether the exclusion criteria were to strict for defining patient target population and whether for example patients who received a dose >50Gy could have been included to improve the robustness of the model.
Finally, I would like to see the authors expand a bit more on what they thought the underlying biology of the identified patients subgroups was - medial LR-PFS of 9 months following definitive CRT seems quite low and I would like to understand the reason for this better.
Overall a very interesting study that addresses several important clinical questions and has the potential to advance clinical practice
Author Response
The study addresses an important question of predicting outcome following CRT for inoperable oesophageal cancer.
This is a retrospective study assessing the potential of PET radiomics to predict the primary endpoint of loco-regional progression free survival (LR-PFS).
The methods are clearly outlined and the in house radiomics extraction software has been validated in accordance with IBSI.
There are a few limitations to this study (that the authors acknowledge) - retrospective nature of the study, small sample size, high risk of overfitting given large number of features examined.
Reply: thank you very much for your comments and your appreciation.
In terms of clinical relevance, I would like to understand how the radiomic based patient stratification compares to clinico-pathologic prognostication (based on histology and TN staging). It would be helpful to either compare performance of clinical model alone with a radiomic or mixed (clinico-radiomic) model to understand the added benefit of radiomic analysis or at least compare significant clinical and pathological variables between two radiomic groups to ensure that the difference in LR-PFS rates are not explained by cervical location, higher T or N stage or SCC vs adeno histology.
Reply: Thanks you for your comment. The clinical characteristics of the 2 groups identified with the heatmap have been compared using a cox univariate analysis in order to verify their impact on LR-PFS. Using this analysis, 2 variables were of potential interest: the initial Charlson Comorbidity index (CCI), with an HR HR=0.46 (95%CI 0.19-1.12; p-value=0.0862) and the T-stage with a HR=2.57 (95%CI 0.95-6.97; pvalue=0.063). As we had only 24 events, we developed a Cox multivariate analysis with these 2 parameters. The following Table shows that at the multivariate analysis they lost any interest.
Anyway, we added these informations in the results of the article.
|
|
n (events) |
HR |
95%CI |
pvalue |
Modele 1 |
46 (24) |
||||
group |
1 |
35 (18) |
1 |
|
0,145 |
|
2 |
11 (6) |
2,16 |
0,77-6,11 |
|
stade T |
T1-2 |
13 (15) |
1 |
0,1612 |
|
|
T3-4 |
33 (19) |
2,1 |
0,74-5,93 |
|
Modele 2. |
46 (24) |
||||
group |
1 |
35 (18) |
1 |
|
0.0469 |
|
2 |
11 (6) |
2.78 |
1.01-7.63 |
|
CCI |
<3 |
16 (8) |
1 |
0.0667 |
|
>=3 |
30 (16) |
0.42 |
0.17-1.06 |
In addition, I wonder whether the exclusion criteria were to strict for defining patient target population and whether for example patients who received a dose >50Gy could have been included to improve the robustness of the model.
Reply: we thank the reviewer for his/her interesting comment. The inclusion of patients treated with higher doses was initially considered when we started this work. But, in such a kind of studies, a lot of parameters (the radiomics parameters) are already analysed. When the number of patients is small, as in our case, the risk of overfitting is really high. So, it is important to reduce as much as possible the number of the other variables. That’s why we decided to exclude the patients as described in Diagram 1, in order to obtain an homogeneous population, and to try to attribute to the radiomics differences the eventual differences in the outcomes of the patients. The impact of the dose on the outcomes of oesophageal cancers patients still remains an open question, and we preferred not to risk to add a variable with a potential impact on the LR-PFS.
Finally, I would like to see the authors expand a bit more on what they thought the underlying biology of the identified patients subgroups was - medial LR-PFS of 9 months following definitive CRT seems quite low and I would like to understand the reason for this better.
Reply: We could agree with the reviewer, but we would like to add some points that could probably make the LR-PFS of not responders patients more “acceptable”.
The group of patient presenting a LR-PFS of 9 months were those presenting worst radiomic features, so a worst prognosis based on the results of our analysis.
Xinopoulos et al (https://pubmed.ncbi.nlm.nih.gov/15610314/) already showed in 2004 that esophageal cancer patients treated with a palliative stent had a median OS of 18 months, and 11/72 patients presented dysphagia because of tumor regrowth in 4-16 week after stenting.
In the INT0126 trial, the 2-year LR-PFS was 52%, ant it is a bit higher than the 32% of our study (see figure 2). In radiochemotherapy group of the RTOG.
In a more recent Phase I/II study, Welsh et al. compared an experimental arm treated with dose escalation, with an historical series treated with 50.4 Gy. In the standard arm, the 2-year LR-PFS was 44%.
Few data exist in the literature exploring the impact of cCR on the clinical outcomes of the patients. A stud by Ahn et al showed that “The complete response rate immediately after radiation therapy was 32% (34/ 106) and the median survival and 2-year survival rate of the complete responder was 14months and 30% respectively while those of the nonresponder was 4 months and 0% respectively with the statistical significance (p=0.000).” Unfortunately, these authors do not give data about LR-PFS, but a similar impact of a worst response on the LR-PFS could be reasonably estimated.
In the study of Xiong et al published in 2018 (Scientific Reports | (2018) 8:9902 | DOI:10.1038/s41598-018-28243-x), exploring the role of radiomics in esophageal cancer, their patient presenting worst radiomic features presented a 2-years LR-PFS of 14.3% (95% CI, 0.0–40.2%), and a median LR-PFS of 6 months.
In the study by Chen et al published in 2019 (Annals of Nuclear Medicine: 33, 657–670 (2019), the DFS of patients presenting bad radiomics features was 10 months.
Looking at these data, it seems to us that the 9 months of LR-PFS of our bad prognosis patients is comparable with available literature.
Overall a very interesting study that addresses several important clinical questions and has the potential to advance clinical practice
Reply: thank you very much for your comments and your appreciation.